# Lipedema: Insights into Morphology, Pathophysiology, and Challenges

**DOI:** 10.3390/biomedicines10123081

**Published:** 2022-11-30

**Authors:** Ankita Poojari, Kapil Dev, Atefeh Rabiee

**Affiliations:** Department of Physiology & Pharmacology, Thomas J. Long School of Pharmacy, University of the Pacific, Stockton, CA 95211, USA

**Keywords:** lipedema, chronic disease, adipose tissue, fat disorder, obesity, lymphedema

## Abstract

Lipedema is an adipofascial disorder that almost exclusively affects women. Lipedema leads to chronic pain, swelling, and other discomforts due to the bilateral and asymmetrical expansion of subcutaneous adipose tissue. Although various distinctive morphological characteristics, such as the hyperproliferation of fat cells, fibrosis, and inflammation, have been characterized in the progression of lipedema, the mechanisms underlying these changes have not yet been fully investigated. In addition, it is challenging to reduce the excessive fat in lipedema patients using conventional weight-loss techniques, such as lifestyle (diet and exercise) changes, bariatric surgery, and pharmacological interventions. Therefore, lipedema patients also go through additional psychosocial distress in the absence of permanent treatment. Research to understand the pathology of lipedema is still in its infancy, but promising markers derived from exosome, cytokine, lipidomic, and metabolomic profiling studies suggest a condition distinct from obesity and lymphedema. Although genetics seems to be a substantial cause of lipedema, due to the small number of patients involved in such studies, the extrapolation of data at a broader scale is challenging. With the current lack of etiology-guided treatments for lipedema, the discovery of new promising biomarkers could provide potential solutions to combat this complex disease. This review aims to address the morphological phenotype of lipedema fat, as well as its unclear pathophysiology, with a primary emphasis on excessive interstitial fluid, extracellular matrix remodeling, and lymphatic and vasculature dysfunction. The potential mechanisms, genetic implications, and proposed biomarkers for lipedema are further discussed in detail. Finally, we mention the challenges related to lipedema and emphasize the prospects of technological interventions to benefit the lipedema community in the future.

## 1. Introduction

In the clinic, lipedema is frequently misdiagnosed as obesity, lymphedema, lipodystrophies, or other fat disorders. It is distinct in its manifestation as solely affecting the upper and lower extremities with a disproportional deposition of subcutaneous fat, sparing the hands and feet [1,2]. Although the diagnosis of lipedema does not exclude the presence of obesity and lymphedema (especially the latter, which is typically present as a comorbidity in the advanced stages of lipedema), it is essential to have a correct diagnosis to manage the symptoms in the early stages of the disease [1,3]. Lipedema patients at an advanced stage experience severe long-term pain and significant psychosocial distress, including depression, eating disorders, and social isolation [4] due to fat-shaming and a lack of adequate medical treatment. Lipedema affects nearly 11% of adult women and postpubertal females globally. As it has been estimated in reports, it could potentially affect millions of adult women in the United States alone, with an incidence of approximately one in nine women [5,6]. Lipedema is characterized by tenderness to palpation, easy bruising, and the bilateral and asymmetrical expansion of inflamed subcutaneous adipose tissue (SAT) [7,8]. The manifestation of lipedema often coincides with periods of hormonal changes that occur throughout puberty, after pregnancy, or during the menopause stage [9] and may be inherited [10]. Although several morphological characteristics are altered in the SAT of lipedema patients, the mechanisms underlying such changes have not yet been clarified. In addition to the nature of the fat, the potential roles of several other factors, such as the immune and lymphatic systems, in the development of lipedema have also been investigated. Unlike obesity, adipocyte hypertrophy and the localized deposition of fat in the SAT of lipedema patients often do not respond to dietary (caloric) restrictions, exercise, and bariatric surgery [4,11,12]. Therefore, further studies to investigate novel biomarkers and mechanistically understand lipedema’s onset and progression are essential. This effort is expected to aid in the timely diagnosis and, eventually, develop treatment modalities toward a better therapeutic outcome for patients with lipedema.

## 2. Lipedema: Morphology

### 2.1. Components of Adipose Tissue and Its Heterogeneity

The health status and function of adipose tissue (AT) depend on the fat-storing adipocytes and the complex intercellular communication between the several cell types residing within the tissue. Heterogenous AT is not solely composed of mature adipocytes but is also composed of adipocyte precursors/stem cells, immune cells, blood cells, and lymphatic capillaries consisting of endothelial cells (ECs) [13,14]. It has become evident that the growth, expansion, and physiological function of white adipose tissue (WAT), a weight-regulated organ, is highly controlled by cross-talk between each of its cellular components, with a central role for the vascular endothelium lining the blood vessels [15]. Microvessels or capillaries within the adipose vasculature are composed of a thin layer of ECs, within which pericytes—a type of supporting cell—are enmeshed, allowing for the efficient delivery of gases, fluids, and essential macromolecules to the parenchyma cells [16]. AT features continuous capillaries with tightly arranged ECs, which are impervious to paracellular leakage and oversee the delivery of nutrients in a trans-endothelial manner (i.e., through the endothelium) through highly controlled mechanisms [17,18]. This continuous endothelial arrangement is accomplished by establishing tight junctions between neighboring cells and a constant membrane along the vessels [15]. It is shown that when an individual AT achieves its maximum growth rate, it loses the ability to store lipids further, resulting in lipid leakage from the tissue, ectopic lipid deposition in peripheral organs, and a systemic deterioration in metabolic health [19]. Although the endothelium was initially thought to be merely a barrier, it has recently emerged as a dynamic unit that regulates many critical functions of AT [15], such as the volume and type of lipids stored in various fat depots [20]. To maintain a healthy microenvironment within the AT, adipocytes are surrounded by interstitial fluid (IF) and are supported by a network of extracellular matrix (ECM) proteins that provide structural integrity [21]. However, as explained above, morphologically, the SAT in lipedema is pathologically altered. Studies on lipedema have included precursor cells and mature adipocytes, as well as impaired adipose vasculature, including the adipose endothelium, the involvement of the IF and lymphatic system, ECM remodeling, and the deposition of collagen, all synchronously promoting fibrosis and edema in the affected tissue [1].

### 2.2. Morphology of Lipedema SAT

A study focused on the morphological assessment of lipedema stem cells derived from the subcutaneous flank. Lateral thigh adipose tissues (tATs) revealed a significant increase in cell number [12]. The increased cell number was accompanied by an elongated, partly spindle-shaped appearance, with rounded nuclei at 11–14 days of culture without adipogenic stimuli. However, on days 7 and 14 of culture with adipogenic stimuli, the cell morphology changed from having a spindle-like appearance to having an adipocyte-like shape [12]. Investigation of the SAT in lipedema patients from the lower extremities showed hypertrophic and hyperplastic adipocytes, increased intercellular fibrosis [7,22], crown-like structures, elevated macrophage levels [7,23], and a morphologically distinct appearance compared to healthy control tissue [22] (Figure 1). It has also been reported that the AT of an individual with lipedema undergoes major structural and functional reprogramming [22], including increased unstimulated lipid release [24], tissue inflammation, and excessive fluid accumulation [6,25].

### 2.3. Excess Interstitial Fluid, Contributing Factors, and Consequences

The swelling caused by adipose hypertrophy occurs in a distinctively symmetrical form in lipedema, which, unlike lymphedema, does not show overt interstitial edema [26]. However, excessive fluid accumulation in the interstitial space is often a common characteristic of progressed lipedema. Increased limb capillary pressure, changes in tissue structure, extra blood vessels with excessive permeability, increased lymphatic area, and inadequate lymphatic outflow are the potential contributing factors that facilitate fluid accretion in lipedema [27].

The increased IF in lipedema allows for the palpation of individual fat lobules as nodules [7]. Moreover, the blood and lymphatic flow have been revealed to be slower in the gynoid SAT in lipedema [28]. Under inadequate flow, inflammatory and fibrotic lesions develop in SAT, leading to chronic pain and palpation [29]. Meanwhile, livedo reticularis and capillary fragility in lipedema LCT can make individuals more prone to bruising [6,30]. In nonlipedema SAT, groups of adipocytes called fat lobules readily glide over one another beneath the skin on tiny wet fibers, giving the skin a uniformly smooth and soft feel [31].

On the other hand, the lipedema tissue exhibits some defining features, such as (1) inflammation that leads to tissue fibrosis in the ECM [23,32] and (2) the formation of palpable fibrotic nodule-like structures within the skin [6]. The aberrant elastic fibers seen in the skin lesions of lipedema patients are thought to be partly caused by progressive mineralization [33]. The loss of smooth consistency and the development of pearl-sized (5 mm) nodules close to the lymph nodes are features of lipedema SAT, which represent tenderness to palpation [1,24]. Ibarra et al. examined nodules that were either within one centimeter of the skin or directly beneath it. These nodules were found to be similar to the palpable lumps felt on examination. Thus, they revealed a connection between blood vessels and hyperechoic masses (~1 cm in size) in lipedema and Dercum’s disease [25]. Notably, such a mass was suggested to indicate a leaky vessel, a bruise, or inflammation around a vessel—all symptoms found in lipedema.

### 2.4. Stages of Lipedema

Lipedema involves fluid in the fat at more advanced stages of the disease. Enlarged blood vessels, skin microangiopathy, and various forms of nodules (i.e., rice-grain, pearl-sized, or larger-sized nodules in LCT) are observed during the progression of lipedema, especially in women [6,34,35,36]. Based on the tissue structure, mobilizing pattern, and pathology, lipedema is characterized by four progressive stages (Table 1). In stage I, the patients present with a smooth skin texture, an enlarged subdermis, pearl-sized nodules in a hypertrophic SAT layer (which are painful once in a while), and a subdermal pebble-like feel due to underlying LCT fibrosis. The following occurs in stage II: skin depressions with pearl- to apple-sized masses that form in the skin, AT, palpable nodules, and bands of perilobular fascia that thicken and contract, an inflamed appearance of the skin (due to progressive fibrotic changes), and pulling down of the skin in a mattress pattern (due to excess tissue). In stage III, patients feature more painful, increased lipedema tissue that is more fibrotic in texture with numerous large subdermal nodules and overhanding lobules of tissue, as well as fat on the arms, hips, thighs, and around the knees. Moreover, in comparison to stage I or stage II, the skin thins and loses elasticity, allowing the SAT to grow excessively and fold over, further inhibiting the flow [37], which can eventually lead to lipolymphedema or lymphedema in stage III [38]. In *stage* IV, lipedema is characterized by lipolymphedema (concomitant lymphedema) which develops in the presence of both lipedema and lymphedema, featuring large overhangs of fat tissue on the legs or arms, and large fat tissue extrusions on the legs that progress to lipolymphedema, thus representing a more advanced stage in most cases [2]. Notably, during all stages of lipedema, lipolymphedema (a condition similar to lymphedema) can occur [39]. For this reason, lipedema is often confused with other metabolic disorders, such as lymphedema, obesity, lipodystrophies, and other fat-related diseases [10]. Therefore, investigating the dysfunctional physiological mechanisms underpinning lipedema is quite challenging for researchers.

## 3. Mechanisms Implicated in the Pathogenesis of Lipedema

As described above, the clinical diagnosis of lipedema relies heavily on the external nature or appearance of the fat in the affected area, with little being known about the underlying etiology of this condition. To date, hormonal imbalances [40], genetic influences [33], an impaired lymphatic system, and vasculature [41] have been frequently suggested as some of the mechanisms underpinning the excessive fat in lipedema (Figure 2).

### 3.1. Altered Gene Expression of Adipogenic and Hormonal Markers in Lipedema 

Preliminary characterization studies have shown the hypertrophic nature of lipedema AT, represented by enlarged adipocytes [36]. Thus, the expression levels of primary adipogenic genes have been assessed by many groups. Table 2 summarizes the genes that have been screened and validated to be linked to lipedema so far. UCP1 is a protein enriched in the inner mitochondrial membrane of metabolically active brown adipose tissue (BAT) with minimal presence in WAT. UCP1 uncouples oxidative phosphorylation from ATP synthesis, leading to the dissipation of biochemical energy as heat [42]. As a lower rate of metabolism in adipocytes leading to hypertrophy has been negatively correlated with the mitochondrial presence, rationally, a lower expression of UCP1 was expected—but not observed—in lipedema AT. Although this study did not find any changes in the expression levels of common adipogenic markers such as leptin (LEP), ADIPOQ, and PPARG in lipedema samples, the expression of CCAT-enhancer-binding protein delta (C/EBPD), nuclear receptor corepressor 2 (NCOR2), and Kruppel-like factor 4 (KLF4)—which are transcription factors (TFs) regulating PPARG expression—were found to be decreased [22]. A 3D culture-based study mimicked the AT microenvironment more suitably than the 2D culture and showed no significant difference in the expression of ADIPOQ, PPARG, and lipoprotein lipase (LPL) among lipedema and healthy controls [36]. Similar findings were obtained by Strohmeier et al., who reported that quantitative polymerase chain reaction (qPCR) analysis showed no significant difference in the expression of LEP, ADIPOQ, and fatty-acid-binding protein 4 (FABP4) in lipedema samples compared to controls [43]. In contradiction with these findings, qPCR gene-expression analysis of differentiated adipose-derived stem cells (ADSCs), obtained from the abdomens and thighs of lipedema patients and healthy lean subjects, showed higher mRNA expression of PPARG and LEP in lipedema ADSCs [44].

The expression of glucose transporter type 4 (GLUT4)—an insulin-sensitive glucose transporter—has been similar in lipedema and healthy ADSCs [44]. GLUT4 expression itself is closely related to the expression of PPARG and CCAT-enhancer-binding protein alpha (C/EBPA), which, under physiological conditions, synergically promote insulin sensitivity [45,46]. Though the stage of lipedema for ADSCs in this study was unknown, these findings seem to align with the fact that the prevalence of diabetes, which relies on insulin sensitivity, is low in lipedema patients [1,47]. The contradictory results regarding PPARG and LEP expression may be explained by the different sources of samples used: in the former study, qPCR analysis was performed using the RNA directly extracted from fat tissue [22], whereas in the latter study, the RNA was extracted from ADSCs differentiated ex vivo using a commercial adipogenic cocktail [44].

Hyperproliferation due to excessive mitotic clonal expansion (MCE) seems to be a common characteristic of lipedema cells [22]. Insulin-like growth factor 1 (IGF-1) levels were significantly higher in undifferentiated ADSCs and lower in mature differentiated adipocytes of lipedema compared to control cells [12]. IGF-1 plays a vital role as an adipokine in regulating the terminal differentiation of 3T3-L1 cells and bypassing the inhibitory action of preadipocyte factor 1 (PREF-1), a soluble factor that maintains cells in an undifferentiated state [48,49]. IGF-1 signaling through mitogen-activated protein kinase (MAPK) promotes MCE in the initial stages of adipogenesis, whereas in the latter stages, it promotes cell-cycle exit and terminal differentiation [50]. Thus, the higher IGF-1 levels noted in the early stages of adipogenesis, followed by the later decreased expression, might contribute to the pathologic phenotype of hyperproliferative adipocytes in lipedema.

Another gene, cyclin D1 (CCND1), has been pinned down as a potential target contributing to the hyperplastic phenotype of lipedema AT due to its positive correlation with MCE [51,52,53]. Following these findings, Strohmeier et al. observed the upregulation of zinc finger protein 423 (ZNF423) in the lipedema stromal vascular fraction (SVF) and lipedema tAT pericytes compared to the respective control groups [43]. ZNF423 plays a crucial role in early preadipocyte commitment by inducing PPARG expression through the bone morphogenetic protein (BMP) signaling axis. The higher expression of ZNF423 may also contribute to the hyperproliferative phenotype of AT in lipedema. The upregulation of ZNF423 could potentially explain the lower thermogenic rate of AT in lipedema fat by maintaining the white phenotype of AT and suppressing the brown–beige characteristics [54].

About adipokines, no differences were found in the serum concentrations of interleukin-6 (IL-6), interleukin-18 (IL-18), lipocalin-2 (LCN-2), and LEP between the lipedema and control samples [22]. Another group [55] sought to investigate the relationship between the common adipokines of adiponectin (ADIPOQ), ghrelin (GHR), resistin (RETN), and visfatin (VISF), as measured by ELISA and SAT thickness (evaluated by ultrasound) in lipedema patients. Similarly, these authors observed no difference in the serum levels of the adipokines, as mentioned above, between the lipedema and control samples, and no significant correlation between the adipokine levels and SAT thickness was identified [55].

The onset of lipedema is closely associated with periods of hormonal fluctuation, such as puberty, pregnancy, or menopause, which have been linked to estrogen hormone/receptor dysregulation. In women, estrogen is associated with subcutaneous lipid accumulation in the femoral and gluteal regions, and the loss of it in menopause leads to the expansion of visceral fat in the abdominal area [56,57]. Gene-expression analysis showed a higher expression of aromatase (CYP19A1)—responsible for the conversion of androgens to estrogen—in lipedema tAT compared to lean healthy control tAT and the abdominal AT of the same lipedema patient. However, estrogen-receptor gene-expression levels have been shown to be unaffected in lipedema [40,43]. In highly proliferative breast cancer cells, estrogen was found to induce ZNF423 expression, a TF that has been confirmed to be upregulated in lipedema [58]. ZNF423 expression in preadipocytes is related to the activation of PPARG and directly to the maturation of adipocytes; thus, upregulation of ZNF423 is a potential mechanism by which estrogen promotes hyperproliferation in lipedema.

### 3.2. Impairment of Endothelial Junction and Increased Permeability of Endothelial Cells in Lipedema

Lipedema is characterized by insufficient backflow leading to excess fluid accumulated in the interstitium. The excess IF might be associated with adipocyte hypertrophy and hyperplasia, inadvertently leading to hypoxia, microangiopathy, and a higher permeability of endothelial capillaries [7,43,59,60]. Microangiopathy in tAT (not the abdomen) for lipedema has been confirmed, which is the reason for the increased permeability of the microvessels leading to excess IF. The extra IF surrounding the cells that are not picked up by the lymphatic vessels is, in turn, in contact with the cellular layers, thus nourishing them. As a result, the dysregulated endothelial function leads to the excessive growth of fat and the pathological remodeling of AT in a cyclic manner [27,61] (Figure 2). Such remodeling, similar to obesity but in a different fat depot, has also been hypothesized as the reason underlying the hypoxic conditions within lipedema AT, leading to necrosis, inflammation, and fibrosis [7,22,62].

Through the immunostaining of junction proteins and the use of a machine-learning approach, the differences in the endothelial junctions between lipedema and control SVF-derived ECs have been investigated [43]. Primary human endothelial cells (hECs) treated with lipedema SVF-derived conditioned media (CM) weakened endothelial junctions, which markedly increased endothelial permeability, as measured by the leaking of the fluorophore NaF (sodium fluorescein solution) through the endothelium of hECs [43]. Analysis of the proteins associated with endothelial barrier integrity, including VE-cadherin, zona occludens-1 (ZO-1), E-selectin, and tunica interna endothelial cell kinase 2 (TIE-2), in hECs treated with lipedema CM was recently performed. A significant decrease in the expression of Cadherin 5/VE-cadherin (CDH5) and a slightly decreased expression of ZO-1 (a component of the endothelial tight junction) was observed, with no difference in the expression of E-selectin and TIE-2 compared to hECs incubated with CM from control cells [43]. However, in another study, the relative gene expression of TIE-2 was downregulated in AT samples of lipedema patients [32]. TIE-2 activation by its ligand, angiopoietin-1 (ANGPT-1), supports vascular maintenance by forming a tightened barrier. The barrier defense induced by ANGPT-1 involves the activation of TIE-2, which reorganizes the actin cytoskeleton and accumulates CDH5 at the endothelial junction [63]. Thus, the results of the latter study are in line with the fact that the functions of CDH5 and TIE-2 seem to be intricately connected, leading to a tightened endothelial junction with marked impermeability [63], where the decreased expression of CDH5 was previously confirmed in a former study [43]. However, the pros and cons exist in both studies. The findings by Strohmeier et al. were derived from hECs exposed to CM from lipedema samples. At the same time, the latter study, carried out by Felmerer et al., was performed directly on dissected fat tissue. As such, the extrapolation of the results to the endothelial level may be a bit of a stretch.

### 3.3. The Altered States of ECM in Lipedema

Lipedema is a connective tissue disorder characterized by a loss of elastic recoil, allowing more fluid to enter the interstitial space and accumulate between dermal skin fibers rather than being cleared by the lymphatic system [1,27].

Nonpitting edema in the early phases of lipedema (the stages of lipedema are described in Table 1) indicates the dissemination of fluid in the tissue’s interstitial space and its attachment to glycosaminoglycans [25]. Sodium and water are usually bound by glycosaminoglycans because of their potent negative charge, and it has been postulated that the concentration of glycosaminoglycans rises when the amounts of salt or water in the ECM increase [6]. In the presence of excess fluid, LCT becomes acquiescent, allowing more fluid to accumulate and promoting proteoglycan production. ECM fluid accumulation, free and bound to proproteoglycans, also increases lymphedema [64], similar to the lipolymphedema representing stage III lipedema (Table 1). In a previous study, the researchers observed an increased number of proteoglycans in the enlarged ECM of lipedema patients [22]. In line with this, excessive AT in obese people represents an increased level of specific proteoglycans [65]. In summary, fluid either leaves the tissue through lymphatic vessels [66] or stays inside the tissue, attached to proteoglycans and glycosaminoglycans. Disturbance to the glycocalyx—which lines all vessels and is made up of proteoglycans, glycoproteins, and related glycosaminoglycans—may cause microangiopathy in lipedema patients [67]. The tissue of lipedema patients has higher sodium concentrations [8], which impair the glycocalyx barrier function of ECs, making them more prone to inflammation [68].

Not only is endothelial permeability compromised in lipedema, but the ECM also undergoes pathological remodeling to make room for hypertrophic and hyperplastic adipocytes. Immunohistochemistry results have confirmed that differentiated spheroids derived from lipedema ADSCs showed higher expression levels of basement membrane components of laminin (LAM) and collagen VI (COL6A3) and lower expression of fibronectin (FN) compared to undifferentiated and healthy spheroids [36]. The elevated expression of both LAM and COL6A3 has previously been linked to the obese phenotype [69,70]. Following high-fat diet (HFD) feeding in mice, the expressions of laminin 2 (LAM2), laminin 4 (LAM4), and several collagen subunits were upregulated. Notably, LAM4 was predominantly upregulated in obese SAT samples at both mRNA and protein levels compared to lean controls [70]. Eight weeks of overfeeding increased COL6A3 mRNA expression, VAT mass, and macrophage infiltration and was linked to a more obese phenotype [71]. The decreased expression of FN has been associated with a reduced differentiation potential of ADSCs in human infrapatellar-fat-pad-derived stem cells (IPFSCs) [72]. Fibronectin-1 (FN-1) knockout has been found to cause the failure of fibronectin fibrillogenesis, a process in which FN undergoes conformational changes that expose fibronectin-binding sites and promote the intermolecular interactions needed for fibril formation, a critical determinant of adipogenesis [72,73]. In lipedema, as a lower expression of FN has been confirmed, failure to undergo fibrillogenesis may be an underlying cause of impaired adipogenesis. Excessive collagen deposition, as a form of ECM remodeling, has also been observed in lipedema [22,36]. The deposition of collagen is under the tight control of matrix metalloproteinases (MMPs), which are expressed similarly in both lipedema and healthy spheroids, except for MMP11, which showed a (nonsignificant) decreased expression [36]. Dysregulation of MMP11 expression was previously established early in AT dysfunction in obesity [74]. Furthermore, the slight decrease in MMP11 expression in lipedema may contribute to excess collagen deposition, as MMP11 is known to cleave COL6A3, eventually reducing collagen deposition [36,75].

An ECM/lymphatic-related signaling mechanism hypothesized for the pathophysiology of lipedema is the matrix metallopeptidase 14 (MMP14)–caveolin-1 (CAV-1) axis, wherein MMP14 and CAV-1 are mutually regulated through a feedback mechanism [76,77]. Overexpression of MMP14 in this proposed pathway has been associated with a hypertrophic phenotype in SAT, decreased expression of the Prospero homeobox 1 (PROX-1) master regulator of the lymphatic system, dysfunction of estrogen signaling, increased permeability, and fragility of blood vessels [77]. Surprisingly, Felmerer et al. did not find any significant difference in the expression of PROX-1 in lipedema fat tissue compared to healthy controls [32].

### 3.4. Immune-Cell Recruitment and Altered Cytokine Profile in Lipedema

Leaky vessels and consequent immune-cell recruitment and inflammation have all been associated with lipedema [27,43]. With regard to the immunophenotype, Priglinger et al. found that the immunophenotypes for SVF isolated from both lipedema and healthy patients were similar. The only surface markers that were elevated in freshly isolated lipedema SVF were CD90 (mesenchymal) and CD146 (endothelial) compared to healthy subjects (12 and 20% higher, respectively). Approximately half of the cells within the SVF were positive for CD90 and CD146 [78]. CD90, which is highly expressed in subcutaneous ADSCs (S-ADSCs), is essential for AKT activation and CCND1 upregulation, thereby regulating cell growth and differentiation. An elevated expression of CD90 might be related to the hyperplasia seen in lipedema [79]. CD146 is known to be an endothelial/pericyte marker that acts as a receptor for angiopoietin-like protein 2 (ANGPTL2) and has been implicated in promoting obesity by enhancing adipogenesis and lipogenesis [78,80]. Pericytes play a role in the development and maintenance of blood-vessel integrity, as well as the control of immune-cell-trafficking across the vessel walls [81,82]. The increased expression of a pericyte marker observed in this study indicated the remodeling of the vasculature at the capillary level, which may be one of the many factors responsible for the lipedema phenotype [78].

Felmerer et al., in two separate studies, noted significant increases in macrophage infiltration in lipedema tissues, namely, CD45- and CD68-positive cells [22,32]. In one of the studies, immune-cell composition analysis was performed due to increased expression of vascular endothelial growth factor receptor 3 (VEGFR-3), a marker well-known to be present in macrophages. They confirmed that the recruited macrophages were of the M2 phenotype due to the overexpression of CD163. This scavenger receptor might indicate the presence of a compromised endothelial barrier, thus demanding the presence of macrophages [32]. This is in opposition to their findings in an earlier report, wherein decreased expression of KLF4 was reported [22]. KLF4, similar to CD163, is a marker of M2-polarized macrophages, as it promotes this phenotype and inhibits M1 polarization [83]. M1 macrophages are usually associated with the inflammatory phenotype, and their higher expression in AT has been reported in obesity [84]. M2-polarized macrophages are associated with angiogenesis in preadipocytes [85], and an overabundance of these macrophages has been reported as a culprit in aberrant fibrogenesis [86]. Both microvessel formations, leading to undesired angiogenesis and fibrosis, are morphological features commonly observed in lipedema. Thus, these paradoxical findings need to be investigated further, as macrophages have been confirmed to be drivers of the inflammatory phenotype seen in lipedema without the involvement of the T-cell compartment [32].

Through qPCR analysis, it has been found that CD11c (ITGAX)—a marker gene for immune-cell infiltration—had significantly higher expression in AT of lipedema tAT compared to the controls [43].

Based on a membrane-based antibody array, levels of selected proteins known to play roles in endothelial function and inflammatory processes have been determined in cell-culture media harvested from the thigh-derived SVF of healthy and lipedema individuals. Only the expression of chemokine IL-8 was found to be significantly decreased in lipedema [43]. ELISA performed on the supernatant derived from lipedema ADSCs indicated a higher concentration of IL-8 than healthy controls. However, under adipogenic stimulation, the secretion of IL-8 decreased in both lipedema and nonlipedema adipocytes, with comparable levels on day seven of differentiation [12]. In the case of lipedema, reduced levels of IL-8—a proinflammatory cytokine—may be explained by higher levels of estrogen. Estrogen-receptor-positive breast cancer cells have been correlated with lower levels of IL-8 (higher levels of IL-8 have also been associated with increased metastasis and angiogenesis) [43,87,88].

With regard to other cytokines, the expression level of VEGFR-3 has been found to increase with the decreased expression of vascular endothelial growth factor A (VEGF-A) and vascular endothelial growth factor D (VEGF-D) in the AT of lipedema patients [32]. According to Felmerer et al., even though VEGF-A and VEGF-D are implicated in the induction of vascular permeability [89,90], these findings, when tied to increased vascular endothelial growth factor C (VEGF-C; potent vascular permeability factor promoting lymphangiogenesis) in the blood sera of lipedema patients and downregulation of TIE-2, paint a picture of a feedback mechanism involved in repairing the impairment of vascular permeability [32].

### 3.5. Vasculature of Lipedema Adipose Tissue

Dysfunctional vasculature could also be one of the factors contributing to excess IF in lipedema tissue. Blood vessels with higher permeability, associated with the enlarged size of the interstitial space in lymphedema, have been previously described [91]. Similarly, blood-vessel capillaries in lipedema have also been revealed to be hyperpermeable [92] and may provide one way that extra fluid enters the interstitial space [25]. In addition, the more dilated blood microvessels in lipedema provide more fluid to the ECM [23]. Moreover, the dilated venules that cause capillary leakage have been hypothesized to cause dysregulated vascular function in lipedema SAT [93], resulting in vascular sclerosis, IF accumulation, and the deposition of fibrotic materials [7].

Early research by Bilancini et al. showed that functional changes in the lymphatic vasculature are consistently linked to lipedema. Dynamic imaging demonstrated that patients with lipedema exhibited an aberrant lymphoscintigraphic pattern caused by a slower lymphatic flow compared to the abnormalities seen in lymphedema patients [94]. The decreased lymphatic outflow in lipedema adds excess fluid to the ECM [95]. Convoluted lymphatic veins in the legs of lipedema patients led to delayed radionuclide transit within the veins, as observed by lymphangioscintigraphy [95]. In addition to being the outcome of AT expansion [96], lymphatic dysregulation may also play a role in the etiology of lipedema.

The lymphatic system plays an indispensable role in immune regulation, the drainage of excess IF, the absorption of dietary fat in the intestine, and lipid homeostasis [97,98]. Under physiological conditions, IF surrounding the AT flows into the lymph capillaries and ends up in the lymphatic vessels. Edema in the early stages of lipedema points to a disability of the lymphatic system to take up the extra fluid [5,99]. Although existing research has confirmed the role of lymphatic dysfunction in lipedema [100,101], it remains unclear whether the excessive proliferation of adipocytes is responsible for lymphatic system impairment or if the impairment itself is an underlying etiology for lipedema, as defects in the lymphatic system is a causative factor of AT expansion [96,102].

Siems et al. observed a twofold increase in VEGF-A expression in the blood samples of lipedema patients, which was implicated in increased angiogenesis and capillary fragility [41]. Moreover, estrogen plays a regulatory role in VEGF-A in AT. In 3T3-L1 cells, 17-beta estradiol (E2) and estrogen receptor 1 (ESR1) agonist 1,3,5-tris(4-hydroxyphenyl)-4-propyl-1H-pyrazole (PPT) induced VEGF-A expression by binding to hypoxia-inducible factor 1 alpha subunit (HIF1A) in the VEGF-A gene promoter [103]. Thus, impairment of estrogen signaling may also be a culprit in the lymphatic dysfunction in lipedema, as has been hypothesized with the MMP14–caveolin-1 (CAV-1) axis mechanism [77].

Levels of lymphangiogenic markers have been evaluated, and increased expression of VEGF-C in the serum of lipedema patients was confirmed. However, contrary to the study of Siems et al., this study did not show any differences in the levels of VEGF-D and VEGF-A among lipedema and control subjects [60]. Further, this increase in VEGF-C expression was not linked to any visible changes at the morphological level for lipedema, which might be due to comparatively lower serum levels of VEGF-C observed in lipedema compared to lymphedema patients. Higher expression of VEGF-C in lymphedema is shown to be linked to lymphatic remodeling [32,104,105]. As noted earlier, an investigation of the expression profile of the most common lymphatic markers revealed that the expression of only VEGFR-3 was elevated (1.9-fold). The expression levels of other lymphatic-related genes, such as podoplanin (PDPN), PROX-1, lymphatic-vessel endothelial hyaluronan receptor 1 (LYVE1), and C-C motif chemokine ligand 21 (CCL21), were similar in lipedema fat tissue compared to healthy controls [32]. Felmerer et al. concluded that there is no involvement of lymphatic system dysfunction in the development of the lipedema phenotype; however, the lymphatic disorder itself may be a consequence of edema [32].

### 3.6. Altered Lipid Composition and Metabolic Phenotype in Lipedema

In ADSCs stimulated to differentiate into adipocytes, the number of fat droplets per cell was significantly increased in lipedema adipocytes compared to the control. Lipidomic analysis by liquid chromatography–mass spectrometry (LC-MS) identified 928 lipid species. Of the 928 identified lipid species, 112 were found to be significantly altered in lipedema. The leading significantly increased lipids in lipedema included glycerophospholipids (GPLs), LPE (24:1), PC (28:2), PC (26:0), PE (42:2), PE (42:1), LysoPC (24:1), and PC [24]. Higher proportions of glycerophospholipids and sphingolipids have been found in lipedema ADSCs, both of which have been implicated in the metabolic dysfunction of AT [106,107]; for example, ceramide synthase 6-derived C16:0 sphingolipids bind to the mitochondrial fission factor (MFF) and induce mitochondrial fragmentation in vitro, promoting obesity [108].

To decode the lipid composition of lipedema adipocytes, Wolf et al. analyzed the oily phase of the lipoaspirate and serum of lipedema patients by mass spectroscopy. They observed a similar lipid profile in both lipedema and control samples, which led them to conclude that lipedema is not a disease of aberrant lipid metabolism [60]; however, they remarked on the apparent difference in the lipid profile observed in lipedema—compared to obesity and lymphedema, lipedema is characterized by decreased lysophosphatidylcholine in plasma [60]. The latter two have higher levels of cyclopropane-type fatty acids and inflammatory mediators in the oil phase of lipoaspirates [109,110]. Through a metabolomics approach, 640 distinct metabolites were identified in lipedema adipocytes, with the highest metabolite classes belonging to amino acid and carbohydrate metabolism. The significant metabolite differences in lipedema, compared to the control, were annotated to be associated with amino acid metabolism (specifically, lysine biosynthesis and glutamate metabolism), peptides, and glucagon-like peptides (GLPs). An altered metabolomic profile has also been confirmed in disorders such as obesity and type 2 diabetes (T2D); as an example, tryptophan oxidation in the AT of obese patients with metabolic syndrome has been shown to be responsible for elevated levels of kynurenine, a precursor of diabetogenic substances that have been associated with T2D [111].

Functional assessment of the mitochondria from lipedema SVF revealed a higher oxygen consumption rate than control SVF, indicating that lipedema SVF has an increased oxidative metabolic capacity. Wolf et al. further reported that an altered metabolic status of macrophages is essential for macrophage polarization and function [60]. In M1 macrophages, aerobic glycolysis is induced upon activation, involving an increase in glucose uptake and the conversion of pyruvate to lactate. At the same time, the activities of the respiratory chain are reduced, allowing for reactive oxygen species production [112]. M2 macrophages, on the other hand, obtain their energy from fatty acid oxidation and oxidative metabolism. Under the IL-4 activation of M2 macrophages, STAT6 induces PPARG coactivator-1B (PGC1B), which further induces mitochondrial respiration and mitochondrial biogenesis.

Furthermore, PGC1B, together with other TFs, nuclear factor 1 (NRF-1), and estrogen-related receptor alpha (ERRa), drives the production of critical mitochondrial components, such as cytochrome c and ATP [113,114]. Thus, PGC1B is a crucial driver of higher oxidative capacity in M2 macrophages and acts as a metabolic switch, with its lack of function leading to the M1 phenotype. Similarly, growth-differentiation factor 15 (GDF15) induces the polarization of macrophages to the M2 phenotype by upregulating the oxidative function [115]. As the infiltration of M2 macrophages has been confirmed in lipedema [32], this higher oxidative metabolic capacity in lipedema SVF (at least partly) correlates with the phenotype macrophages present in lipedema [60].

AT is a dynamic organ, and apart from storing energy, it also maintains overall homeostasis by secreting lipids, adipokines, extracellular vesicles (EVs), and micro-RNAs [78]. By functioning in an autocrine, paracrine, and systemic manner, these factors contribute to overall metabolic health. Thus, to confirm whether an altered miRNA profile drives the pathogenesis of lipedema or whether lipedema itself is the result of an altered secretome profile, the extracellular micro-RNAs from the SVF of healthy and lipedema ATs have been analyzed and compared. A total of 187 extracellular miRNAs were analyzed in a concentrated conditioned medium (cCM) and in small extracellular vesicles (sEVs), and most of the variant miRNAs showed up in sEV profiles but not in cCM profiles. Primary sEV miRNAs (miR-16-5p, miR-29a-3p, miR-24-3p, miR-454-p, miR-130-3p, and let-7c-5p) were differently expressed in lipedema compared to only one (miR-188-5p) in cCM [78]. However, some of these micro-RNAs have already been implicated in various dysregulations of AT (Table 2), and thus are not unique or specific to lipedema. The critical point identified in the study is that, for lipedema, miRNAs differentiating lipedema and a healthy state exist at the sEV level, pointing to the fact that altered miRNAs of EVs are essential for lipedema rather than the total RNA.

Given the distinct nature of lipedema AT, both in its morphology and gene expression, impairment concerning metabolic function causing the pathogenesis of lipedema is undoubtedly a possibility.

### 3.7. Role of Selenium and Sodium in the Pathology of Lipedema

Siems et al. showed that, in lipedema patients, the serum concentrations of oxidative markers, such as malondialdehyde (MDA) and protein carbonyls, were higher than healthy controls [41]. Selenium has been found to have antioxidant effects and a negative effect on inflammatory markers in AT [116,117]. A deficiency of selenium disrupts the normal functioning of immune cells, such as the disruption of protein folding and calcium flux [116]. Selenium deficiency has been observed in lipedema patients, but this deficiency was much higher in patients suffering from lymphedema, lipolymphedema (both lipedema and lymphedema), and obesity [118]. Furthermore, selenium deficiency has been suggested as a suitable marker to differentiate between lymphedema and lipedema, as lipedema did not present decreasing selenium levels with increasing stages of the disease.

The tissue sodium content was significantly elevated in the skin and SAT in lipedema patients compared to the control. Crescenzi et al. stated that the clearance capacity of lymphatic capillaries is impaired when sodium accumulates in the interstitium [8]. As lymphedema often coexists with lipedema, especially in the later stages, increased sodium levels directly limiting lymphatic clearance might be a possible mechanism for edema formation. Furthermore, elevated sodium levels may mainly attract salt-sensing macrophages, which, through the release of inflammatory mediators, may play a role in the pathology of lipedema [67].

The glycocalyx is composed of proteoglycans, glycoproteins, and associated glycosaminoglycans that line microvessels and act as barriers to prevent the entry of foreign pathogens [68]. Herbst et al. hypothesized that the microangiopathy seen in lipedema is a result of the impaired glycocalyx barrier function of ECs due to elevated sodium levels [67], putting the endothelium at risk of inflammation.

In another study, researchers observed increased cerebral blood flow (CBF) in participants with lipedema [119]. They attributed this to several factors, including (but not limited to) estrogen signaling, wherein higher estrogen levels have been linked to increased carotid arterial blood flow [120].

The literature on lipedema seems to indicate a complex, multifactorial condition distinct from other fat disorders, as shown by the cytokine and lipid profiles of SAT. In lipedema, many of the implicated immunogenic and TFs are (directly or indirectly) regulated by estrogen. Several reported findings are contradictory, especially in earlier reports, due to variations in the sample sources (e.g., SVF, AT, serum, ADSCs, and spheroids derived from ADSCs) and the stage(s) of lipedema being studied.

## 4. Genetic Implications

Despite the limited knowledge about the genes and proteins associated with lipedema or detectable biomarkers in the serum of lipedema patients, inheritance has been reported in 60% of patients, wherein the percentage may likely be higher due to underdiagnosis [4]. In six families over three generations, the inheritance pattern was found to be autosomal dominant with incomplete penetrance, meaning that the pathogenic trait is not expressed in all individuals carrying the pathogenic variant [4,10]. Inherited lipedema is classified as nonsyndromic, syndromic, and/or associated with comorbidities [33] (Figure 3).

For nonsyndromic lipedema (when lipedema is not a component of a spectrum of other conditions existing in the patients and not all clinical features associated with lipedema are manifested), genetic inheritance has been reported without any causative gene/genes being implicated. Whole-exome sequencing was performed for a family with monogenic nonsyndromic lipedema, wherein variants were found to reside in genes that play a regulatory role in steroid hormone signaling [121]. Neuronal guanine nucleotide exchange factor (NGEF), with its known role linked to abdominal obesity, and F-box/LRR-repeat protein 7 (FBXL7), linked to metabolic syndrome and a modified response to corticosteroids, were among the genes found to be altered. However, the highlight of this study was a variant (c.638T > A; p. Leu213G1n) in aldo-keto reductase family 1 member C1 (AKR1C1), confirmed in the SAT of three affected females of the family [121]. AKR1C1 is a gene involved in progesterone metabolism, and higher expression has been established in the SAT of obese women [122].

The role of sex hormones in the pathogenesis of lipedema cannot be stressed enough. Although impairment in estrogen signaling/receptors is often highlighted [40], both isoforms of progesterone receptors have also been confirmed in human SAT [123]. In rats, progesterone has been reported to nullify the weight-reducing effects of estradiol [121,124], and, in women, higher progesterone levels in pregnancy correlated with weight gain [125]. In isolated mature adipocytes, progesterone is converted into its inactive form, 20-α-hydroxyprogesterone, by AKR1C1, thus playing a role in the fat accumulation of the subcutaneous depot [122]. Michelini et al. concluded a partial loss-of-function of AKR1C1 with a negative effect on the conversion of progesterone to 20-α-hydroxyprogesterone, thus potentially contributing to more weight gain [121].

Syndromic lipedema is when lipedema coexists with other disease conditions. Understanding lipedema in the presence of other conditions may provide insight into the altered genes, which may also be of interest for nonsyndromic lipedema in general.

Lipedema has been confirmed in four generations of a family wherein lipedema, short stature, multiple pituitary hormone deficiencies, secondary hypothyroidism, and hyperprolactinemia were present (affected males had short stature and no lipedema). In this study, a mutation was identified in pituitary-specific positive transcription factor 1 (PIT-1), 196C > T, which produces the amino acid change P24L in exon 1. PIT-1 mutation has been linked to many symptoms manifested in pituitary hormonal deficiencies and lipedema [126]. PIT-1 is a TF localized to the anterior pituitary gland, which plays a regulatory role in expressing growth hormone (GH), prolactin, and thyroid hormone beta-subunit genes [33,127]. GH deficiency has been linked to an increase in fat mass, and GH treatment in children and adults with GH deficiency reduces abdominal fat mass and decreases fat-cell size [126]. The secretion of GH is also sex-dependent, with three times higher secretion in women compared to men. This sex-dependent difference in the GH level is related to estrogen, as providing recombinant human GH to GH-deficient adults requires a significantly higher dose for females than males [126,128]. This has been shown in rat hepatocytes where, after estrogen administration, a decrease in the expression of the GH receptor gene was confirmed [129].

Sotos syndrome is characterized by pre- and postnatal overgrowth, macrocephaly, typical facial gestalt, large hands and feet, accelerated skeletal age, and developmental delay. Around 50–75% of Sotos syndrome patients demonstrate mutations in the nuclear-receptor-binding SET domain protein 1 (NSD1) gene or a microdeletion at chromosome 5q35. In a familial case of Sotos syndrome (mother and two children, a son and daughter affected), it was found that a novel NSD1 mutation, C2175S, also caused the condition. The mother, along with the above-listed symptoms, was also diagnosed with lipedema [33,130]. In this report, the NSD1-specific mutation was not directly linked to lipedema; however, NSD1 has previously been reported to encode a protein that, in conjunction with other coregulators, binds to retinoid X, estrogen, and thyroid receptors, wherein it could either act as a co-repressor or co-activator of the nuclear receptors depending on the cellular context and presence/absence of the respective hormones [127,131,132].

Individuals afflicted with Williams syndrome (WS) have been characterized as having altered body composition, lipedema, and decreased bone density [133]. Several features make the lipedema phenotype observed in WS distinct, including a higher prevalence (approx. 20% of patients affected), predominant occurrence in men, and the absence of pain, tenderness, and easy bruising of the problematic areas in persons with WS [133]. No genetic factors linking lipedema and WS have been reported.

A next-generation sequencing (NGS) panel comprising 305 genes strongly linked to lipedema and/or overlapping diseases relevant to lipedema has been developed. Genome sequencing of 162 Italian and American patients with lipedema was performed, and 21 heterozygous detrimental variants were detected, according to 3 out of 5 predictors, including perilipin 1 (PLIN1), lipase E (LIPE), aldehyde dehydrogenase 18 family member A1 (ALDH18A1), PPARG, growth hormone receptor (GHR), INSR, ryanodine receptor 1 (RYR1), Niemann–pick C intracellular cholesterol transporter 1 (NPC1), proopiomelanocortin (POMC), nuclear receptor subfamily 0 group B member 2 (NR0B2), glucokinase regulator (GCKR), and peroxisome proliferator-activated receptor alpha (PPARA) in 17 patients [134]. Variations of the above-listed genes (except for GHR and ALDH18A1) have been linked to several fat disorders. Meanwhile, variants in ALDH18A1 have been linked to connective tissue disorders known as cutis laxa, wherein overhangs of skin folds are observed due to a decrease in elastic tissue formation, similar to the hypermobile joints observed in lipedema [134]. Mutations of GHR have been linked to partial GH deficiency and short stature, which have been confirmed in a single familial case of lipedema [126]. A common trend in these variants is their involvement in steroid biosynthesis, lipid metabolism, and insulin signaling [134]. Knowing genetic variants or polymorphisms associated with genes of interest in lipedema through NGS will aid concurrent analyses of multiple genes in large cohorts of patients, helping develop diagnostic parameters/therapeutic options for lipedema.

Genome-wide single-nucleotide polymorphism (SNP) genotyping of white British lipedema patients revealed a strong genetic link to the disorder, including loci associated with hormone biosynthesis, lipid hydroxylation, and lipoma formation. The three primary SNPs (rs1409440, rs7994616, and rs11616618) were confirmed to be associated with LHFPL tetraspan subfamily member 6 (LHFPL6) expression [135]. In previous reports, LHFPL6 has been linked to a translocation-associated lipoma. An association of lipedema to carboxypeptidase E protein (CPE) has also been confirmed by a genome-wide association study (GWAS). CPE is involved in the biosynthesis of many neuropeptides and peptide hormones, including estrogen [135]. Adipose tissue is a secretory organ, which plays an endocrine role in releasing proinflammatory cytokines, including IL-6. Elevated levels of IL-6 and C-reactive protein have been linked to obesity, which itself represents a low-grade chronic inflammatory state [136,137]. Several polymorphisms in the 5’ region of the IL-6 gene have been identified, but most of the focus has been limited to G/C substitution at position 174 [138]. A specific IL-6 polymorphism (rs1800795) has been found to affect the transcription of IL-6, which has been implicated in weight gain and increased insulin sensitivity. A significant difference in fat distribution was found for carriers of IL-6 polymorphism (rs1800795) vs. noncarriers. In addition, being a carrier of the mutation increased the risk of developing lipedema by approximately six times. A shortcoming of this study was the limited number of patients and the lack of ethnic diversity in the cohort (only the Italian population was considered) [138].

A comprehensive omics-based study recently revealed 4391 significant differences in the mRNA expression of genes involved in critical signaling pathways, such as lipid metabolism and proliferation/cell cycle, in lipedema tissue. The gene-expression profile of this study favored adipose hyperproliferation, fibrosis, and inflammation, which are the distinctive features observed in lipedema [24]. The higher proliferation rate of lipedema ADSCs (lipedema AT had a higher number of precursor cells identified by CD29+/CD34+ expression) was linked to an increased expression level of genes involved in cell-cycle regulation and proliferation [24]. Among the genes related to cell mitotic pathways regulating the spindle-assembly checkpoint, whose expressions were altered in lipedema ADSCs, BUB1 was found to be of particular interest. As a mitotic checkpoint serine/threonine kinase, BUB1 overexpression has been associated with increased cellular proliferation and has been implicated in several cancers [139,140,141]. With regard to lipedema, it was postulated that, by increasing the phosphorylation of histone H2A, BUB1 interrupts the checkpoint between the G1 to the S phase of the cell cycle, leading to an increased number of cells in the S/synthesis phase, followed by decreased expression of PPARG, thereby causing a delay in the process of differentiation [142]. The role of BUB1 in excessive proliferation has not only been implicated in cancer but also in 3T3-L1 cells, where steroid receptor RNA activator (SRA) plays a modulatory role in assisting withdrawal from the cell cycle during the terminal stages of adipogenesis. Its overexpression led to more than a 2-fold downregulation of the BUB1 gene [143].

The fate of preadipocytes towards terminal differentiation and cell-cycle re-entry is exclusively determined in the G1 phase by stimulating a PPARG-driven switch inducing the higher expression and increased stability of the p21 cyclin-dependent kinase (CDK) inhibitor, resulting in permanent cell-cycle exit. The p21 and cyclin D levels, with their opposing roles (adipogenic vs. mitogenic, respectively), determine whether a cell goes through another cell cycle or terminally differentiates (termed G1 competition). As the expressions of PPARG and p21 are positively correlated, an increase in the expression of PPARG induces p21 expression, which stops the cells from re-entering further cell cycles [142]. In lipedema fat tissue, there is a disruption at a checkpoint stage between the G1 and S phases due to the elevated expression of BUB1 and increased expression of cyclin D [22,24]. Thus, increased cyclin D expression in the G1 phase of lipedema cells may cause them to continue committing themselves to further cell cycles, leading to the hyperproliferative characteristic of lipedema fat cells.

Genetics are known to play a role in lipedema; however, such a role is still poorly understood. Considering the advancements in screening techniques, such as GWAS and NGS, significant strides have been made to identify variants and SNPs contributing to the inheritance of lipedema. However, the crux of the situation is still the lack of more markers, such as AKR1C1, that can distinguish the lipedema condition from other lipid disorders, not only in one family, but at a much larger scale, which will warrant the use of routine genetic screening for diagnostic purposes.

## 5. Biomarkers: Current and Future Perspectives to Combat Lipedema

Biomarkers have specific characteristics that can be potentially used for diagnosing the disease. Due to misdiagnosis and a lack of development within the field, biomarkers for lipedema have remained elusive. From the current standpoint, significant findings for the establishment of solid imaging standards and the discovery of a few promising biomarkers may provide potential solutions to combat lipedema, namely by contributing to the correct and timely diagnosis of lipedema. A few likely candidates proposed as biomarkers for lipedema are summarized below (Figure 4).

### 5.1. Platelet Factor 4

At present, under a shortfall of novel biomarker-based diagnostic kits, the conclusive diagnosis of lipedema at an early stage is challenging. Researchers have not yet discovered any reliable biomarkers in lipedema, except for the recently identified platelet factor 4 (PF4/CXCL4). Ma et al. analyzed and compared blood-plasma-derived circulating exosomes in mice models and patients with and without known lymphatic pathologies. They discovered that PF4—a plasma-circulating exosomal signature protein—may be employed as a promising novel biomarker in clinical settings to diagnose lymphatic vasculature failure, allowing for the differentiation of lipedema and lymphedema from obesity [144]. Furthermore, they found that lipedema patients had higher PF4 concentrations in blood-plasma-derived circulating exosomes, and there have been recent corroborating arguments that some of the underlying characteristics of this disease are a result of lymphatic abnormalities [144]. Similar to clinical observations in humans, the expression of PF4 was likewise elevated in both young and old PROX-1+/− mice (before and after the onset of obesity) but not in ob/ob mice (nonlymphatic promoted obesity). These results suggest that PF4 might be a new biomarker for lymphatic diseases, including lipedema [144]. PF4 is secreted by platelets upon activation during wound healing and inflammation. Even though the study did not specify the cell type that produced exosomal PF4 or the mechanism driving exosomal PF4 release, it is conceivable that the deteriorating lymphatic system is in charge of mediating such signaling. Earlier evidence indicated that PF4 downregulates proteins associated with the tight junction and increases cell permeability [145]. These studies support the hypothesis that PF4 may play a potential role in lymphatic disorders, making the blood vessels more permeable and decreasing lymphangiogenesis. PF4 also increases T helper 2 (Th2) cytokines and causes the chemotaxis of T cells in a CXCR3-dependent manner [146]. This is important, as it is well-known that T cells (especially Th2 cells) penetrate lymphedematous tissue and contribute to inflammation, fibrosis, and lymphangiogenesis [146,147]. Inhibition of Th2 differentiation reduces fibrosis, enhances lymphatic function, and slows lymphedema progression [147]. Along with imaging evidence provided by other authors [95,101], the results of Ma et al. support the previous hypothesis that lymphatic dysfunction contributes to the etiology of lipedema [144]. As lean individuals and obese patients did not show a significant difference in PF4 levels, this indicates that PF4 is a viable biomarker for differentiation between healthy participants and those with lymphatic abnormalities, regardless of obesity. Although no report has been published on the use of this marker clinically, PF4 could help clinicians to identify lipedema. In summary, searching for additional ideal biomarkers with promising results could provide a more effective way to combat this painful disease.

### 5.2. Sodium

An MRI-based study revealed that lipedema patients had a significantly higher sodium content in the skin and muscle tissue, along with a higher amount of fat in their lower extremities but not their upper extremities, compared to BMI-matched controls [8]. Additionally, compared to controls, individuals with lipedema had considerably varied and disproportionate fat levels in their lower vs. upper extremities [8]. When tissue sodium and fat levels are considered, clinical-strength MRI may evaluate objective diagnostic criteria to noninvasively differentiate lipedema from obesity. As Crescenzi et al. suggested, the role of tissue sodium in the etiology of lipedema should be examined in future studies, as the sodium content of tissues increases with the severity and discomfort of the condition. To maintain sodium homeostasis, the venous and lymphatic systems must keep the interstitium and capillary bed’s plasma sodium concentrations constant [148]. If these mechanisms are dysregulated, the accretion of sodium may occur over time, which could contribute to the progression of the disease. It is well-known that gravity and hydrostatic pressure also work against venolymphatic circulation. The author observed higher sodium levels in the lower extremities consistent with such a gravity dependence [8]. It has been shown that a higher tissue sodium content is related to the clinical signs of inflammation and pain and the etiology of the lipedema stages [8]. Although significant changes in tissue sodium have been shown in several inflammatory and painful disorders, these associations suggest a dysregulation of tissue sodium that may be caused by insufficient vascular clearance or inflammation, consistent with the progression of symptoms in more advanced lipedema stages [8,38]. Lower extremity pain in patients with lipedema may be correlated with elevated tissue sodium. While the precise cause of pain in lipedema is still unknown, central sensitization [149], joint pain and nociceptive pain [1], and autonomic peripheral neuropathies [4] have been hypothesized, which should be further investigated [8].

### 5.3. Extracellular Vesicles and Their Contents

Many molecules released by AT originate from nonadipocyte cells, such as endothelial and immune cells, present in the SVF of AT [150,151]. Nevertheless, most studies have suggested that most extracellular miRNAs are found in protein complexes rather than exosomes/microvesicles [152,153]. The molecular load of EVs reflects the pathophysiological status of their original cells and, thus, can serve as a diagnostic tool. In addition to the cell-type-specific proteins, lipids, and different classes of nucleic acids, EV-contained miRNAs have been found to be potent regulators in biological processes and various diseases [154,155]. Numerous studies have already shown that the EV composition can distinguish health and illness, thus recommending potential diagnostic biomarkers in diabetes, obesity, atherosclerosis, neurodegenerative diseases [156], different cancers [157,158,159], and renal damage [160]. Recent clinical studies have revealed EVs as biomarkers of prostate cancer (Exosome Diagnostics, 2015), Parkinson’s disease (NCT01860118, 2016), and difficult-to-treat arterial hypertension (NCT03034265, 2017). Exosomes are small, endocytic-derived vesicles (30–100 nm in diameter) released by most cells, including ECs [161,162]. These EVs include DNA, mRNAs, miRNAs and proteins unique to particular cell types. Additionally, they can have a functional influence once they have been absorbed by recipient cells, making them special intercellular communication mediators [163]. To support the relevance of SVF cells in lipedema and the potential link between extracellular miRNA profiles and disease phenotypes, Priglinger et al. systematically characterized the extracellular miRNA fractions produced by the SVF in lipedema patients and healthy individuals. In a previous study, the adipogenesis, cell volume, and cellular subtype composition of SVF cells from diseased sWAT were altered [59]. Priglinger et al. recently observed that the SVF-derived small EVs exhibited the most various characteristic miRNA profile (i.e., miR–16-5p, miR-24-3p, miR-29a-3p, miR-130a-3p, miR-454-p, miR–144-5p, and let-7c-5p) in lipedema patients. The expressions of the above-mentioned small EV miRNAs were found to be differently regulated. In a concentrated conditioned medium (cCM), cCMmiRNA (miR-188-5p) was observed to be significantly downregulated in lipedema [78].

### 5.4. Lipid Profile

In contrast with obesity or lymphedema, lipedema manifests as a distinct pathological characteristic with particular tissue architecture and nonstandard lipid metabolism. Significantly higher values (upper physiological to pathological ranges) of serum lipid markers, such as total cholesterol (1.31-fold), triglycerides (1.49-fold), and low-density lipoprotein (LDL; 1.46-fold), as well as ApoB (1.37-fold), have been observed in lipedema patients compared to controls [22]. The levels of LDL and ApoB are directly correlated, wherein the ratio of LDL/ApoB is related to the LDL particle diameter (as ApoB is a carrier of LDL, higher values usually signify a risk for cardiovascular disease) [164]; thus, it was no surprise that an increase in ApoB values was found in lipedema serum [22]. High-density lipoprotein (HDL) and associated ApoA were comparable between lipedema and control groups. As the tested serum was primarily obtained from lipedema patients in stage III of the disease (seven vs. two in stage II and one in stage I), these findings mostly confirm aberrant lipid metabolism in the advanced stages of lipedema.

### 5.5. Cytokines

A multiplex immunoassay was performed for 37 cytokines using serum samples from lipedema and control groups, and 22 cytokines were found in lipedema serum. IL-11, IL-28 A, and IL-29 were increased in the sera of lipedema patients compared to the control. IL-29 has been implicated in the pathogenesis of obesity-induced inflammation and insulin resistance by upregulating the expression of IL-8, interleukin-1 beta (IL-1β), and monocyte chemoattractant protein-1 (MCP-1) [165], and both IL-28A and IL-29 are known to be secreted by macrophages [165,166]. IL-11 has been shown to promote cellular proliferation in ADSCs and inhibit adipogenesis in a murine fibroblast cell line, which may account for the hyperplasia seen in lipedema ADSCs [167,168].

## 6. Potential Techniques to Study Lipedema 

Several attempts have been made to use more advanced technologies and approaches to reveal the molecular mechanisms underlying lipedema. Such attempts include the development of 3D spheroids using ADSCs to mimic the microenvironment within AT better and using conditioned media from lipedema ADSCs to treat ECs in a transwell assay investigating endothelial permeability [36,43]. Three-dimensional spheroids using ADSCs from lipedema patients have been developed using a scaffold-based technique. However, how this method compares against other known scaffold and scaffold-free techniques of 3D spheroid formation has not yet been investigated. Thus, the development of techniques enabling quicker turnaround time in screening (e.g., hydrogels with an embossed surface for the mass production of human ADSC-derived spheroids [169]) or other screening models developed by 3D printing (e.g., adipose microtissue-on-chip [170]) are essential to ensure a smooth flow driving drug discovery.

Furthermore, high-throughput omics strategies, such as transcriptomic, lipidomic, and metabolomic analyses, have yielded significantly altered profiles for lipedema tissue compared to healthy controls, especially including the genes involved in cell-cycle/proliferation pathways [24]. As leaky vessels leading to edema have been implicated in the pathogenesis of lipedema, to investigate endothelial junction integrity in an unbiased manner, a machine-learning approach using convolutional neuronal network analysis has been utilized [43]. Additionally, the oxidative metabolic capacity for SVF obtained from lipedema patients has been determined using the seahorse technique, which applies specialized sensors to measure the oxygen consumption rate (OCR) [43].

Single-cell RNA sequencing (sc-RNA-seq) enables the screening of transcriptomes of tens of thousands of cells in a single experiment, allowing for the unbiased analysis of cellular heterogeneity, as well as deciphering the cellular population and the molecular mechanisms underlying disease conditions [171,172]. Thus far, RNA-seq analyses for lipedema ADSCs have yielded noticeable differences in the transcript signature. This includes changes in the expression of genes linked to the cell cycle and division, specifically relating to mitotic spindle checkpoint pathways. However, the elucidation of other facets, such as cellular heterogeneity, changes in cell-to-cell communication, and lineage-specific regulatory changes at a single-cell level, has not yet been conducted for lipedema. Such approaches will potentially reveal the candidate cell populations to be targeted in lipedema treatment.

Despite previous and ongoing efforts regarding lipedema, the daily improvements in adipose biology, metabolic diseases, and obesity—a few are discussed below—are yet to be incorporated into the lipedema field.

As AT is a dynamic organ with a heterogenous cell population, 2D histopathology has certain limitations with regard to providing an ideal overall spatial representation [173,174]. Different methods for 3D histological staining have been developed, among which one study has reported using nanobodies. Nanobodies have been developed and validated against human endothelial cell-selective adhesion molecules (hESAMs), enabling the multidimensional visualization of blood vascular networks [174]. Standard IgG antibodies have several shortcomings, which have been overcome using nanobodies (i.e., allowing for a faster tissue penetration rate and higher solubility). Another study reported the use of the 3D visualization of WAT using whole-mount staining termed iDISCO+ (immunolabeling-enabled three-dimensional imaging of solvent-cleared organs), which involved clearing tissue opaqueness using organic reagents and making the AT transparent, ensuring a more accurate representation of the tissue anatomy, especially with regard to blood vessels and neural fibers [173]. As the vasculature is compromised in lipedema, such techniques might prove helpful in visualizing the alterations to the AT vascular network in lipedema.

As transcriptional profiling has found BUB1 to be an essential target regulating cellular proliferation in lipedema ADSCs, researchers have used a CRISPR/Cas-9 lentiviral system to knock down BUB1, which decreases the proliferation of lipedema ADSCs [24]. CRISPR/Cas-9 technology has applications in the validation of specific genes and may also be used for the therapy of lipedema itself. For example, the knockdown of breast-cancer-associated gene 1 (BRCA-1) by CRISPR/Cas-9 in human ADSCs led to more aggressive behavior in breast cancer cells than wild-type ADSCs due to increased cytokine production [175]. Nuclear receptor-interacting protein 1 (NRIP-1) suppresses UCP1 expression and fatty acid oxidation. In a study, NRIP-1 was disrupted using CRISPR/Cas9 in progenitor cells of human SAT and transplanted in NSG mice. Compared to the control, these mice, after HFD, showed improved glucose tolerance over three weeks, thus showing the advantages of CRISPR/Cas9 and cellular therapy for treating metabolic diseases [176].

Hill et al. reported a Western blotting alternative technique, called capillary western immunoassay (WES), that uses individual capillaries to automate protein loading, separation by size, blocking, washing, and detection, thus improving reproducibility and reliability. Using this technique, Hill et al. validated 10 of 11 antibodies targeting estrogen synthesis and signaling protein in SAT and aimed to use the methodology for lipedema samples [177]. However, as WES requires proprietary reagents and equipment, the utility of this technique beyond a limited number of laboratories is questionable.

## 7. Diagnostic Tools and Clinical Studies

Lymphoscintigraphy is the most commonly used diagnostic approach for lymphedema. Other less-invasive options, such as magnetic-resonance imaging (MRI) or computed tomography, have recently been suggested to differentiate lymphedema from lipedema; however, their potential management applications have not yet been well studied [101,178]. MRI, which is thought to be more beneficial due to its better sensitivity, can also assist in differentiating these two conditions. On the other hand, in patients with lipedema, MRI and computed tomography scans show diffuse fatty hypertrophy over the bilateral lower legs without skin abnormalities. In this case, it might be challenging to distinguish lipedema from other disorders that result in an excess of fatty tissue in the lower extremities. Some of the defining features of lipedema, such as the step-off at the ankles, ease of bruising, pain due to pressure, and inflamed tissue, are considered when making the diagnosis [10]. Hypothetically, some studies have also suggested using an ultrasound approach as an accessible alternative to MRI for examining subcutaneous fat accumulation and distinguishing lipedema from obese patients.

Complications linked to increasingly prevalent functional and cosmetic conditions can be avoided through early lipedema diagnosis and treatment [179]. Hence, thorough investigations with longer-term follow-ups are needed to discover the most promising treatment options. In the current scenario, the main obstacle to investigating the lymphatic- and vascular-related problems in lipedema patients and addressing the unmet clinical requirements for patients with lipedema more widely, in terms of evidence-based treatments, is that noninvasive lymphatic imaging procedures are traditionally underdeveloped. To overcome this obstacle, non-tracer-based MR lymphangiography is recommended as a viable method, showing symptoms of elevated atrial and lymphatic insufficiency in the lower extremities of patients suffering from lipedema. By performing a whole-body fat and water MRI, lipedema was characterized by a 42% lower extremity SAT deposition compared to BMI-matched females without lipedema.

Moreover, some diagnostic tools exist that can be used to examine the limb extracellular water (by bioimpedance spectroscopy), skin water tissue (by the dielectric probe), and skin elasticity by fibrometer; https://clinicaltrials.gov/ct2/show/NCT05464927 (accessed on 29 October 2022). Pneumatic compression devices can be used to stimulate lymphatic flow [180], which can also be used as an option for at-home lipedema and lymphedema management if no contraindications exist [180,181]. In addition to reducing pain, pneumatic compression devices may be more effective at reducing swelling than self-manual lymphatic drainage [182]. Moreover, the risk of deep venous thromboembolism after lipedema reduction surgery can be reduced with its use and prompt mobilization [183]. However, it must be noted that these devices are not a permanent solution to combat lipedema.

## 8. Current Challenges with Lipedema

Presently, lipedema is underdiagnosed and often misinterpreted as other similarly reflecting pathological conditions, such as lymphedema, obesity, Dercum’s disease, multiple symmetric lipomatosis (MSL), and familial multiple lipomatosis. Due to a lack of knowledge on the pathophysiology of lipedema and the numerous unsolved questions regarding its ideal therapeutic management, unfortunately, treatment remains somewhat limited [7,37,179,184]. Studies on lipedema have been limited and constrained by the relatively small number of patients due to the ethical limitations associated with obtaining ideal tissue samples from patients undergoing surgery. To balance out the small sample size, research was previously carried out to examine anatomically similar tissue biopsies, such as skin, serum probe, and fat samples. Similar to lipedema, localized adiposity in some areas of the legs—especially the outer thighs, around the knees, and the lower calves—can be highly resistant to diet and exercise in some women. Further, some women with lipedema may also have additional areas of localized adiposity [94], making the diagnosis even more challenging. As there have not been many randomized controlled trials addressing lipedema as a social issue, research has tended to rely on patient-completed surveys, which are not objective [185]. Therefore, additional research should be conducted to advance our understanding of lipedema; create specific diagnostic criteria; and explore the relationships between lipedema, lifestyle, comorbidities, and quality—all of which shorten the time to receive a diagnosis. Alleviating the lower extremity symptoms and functional limitations associated with lipedema and preventing the progression of the disease are the main objectives of lipedema management. As etiology-guided lipedema management is not yet available, an effective treatment strategy must also consider conditions such as venous or lymphatic edema, decreased physical activity, obesity, and other abnormal conditions that may negatively affect lipedema [186].

Lymphoscintigraphy has been significantly used to diagnose lymphedema. In this line, other, less-invasive options, such as magnetic-resonance imaging (MRI) or computed tomography, have recently been suggested to differentiate lymphedema from lipedema; however, their potential management applications have not yet been well studied [101,178]. MRI, which is thought to be more beneficial due to its better sensitivity, can also assist in differentiating these two conditions. On the other hand, in patients with lipedema, MRI and computed tomography scans show diffuse fatty hypertrophy over the bilateral lower legs without skin abnormalities. As such, it might be challenging to distinguish lipedema from other disorders that result in an excess of adipose tissue in the lower extremities. Some of the defining features of lipedema, such as the step-off at the ankles, ease of bruising, pain due to pressure, inflamed tissue, and so on, are used to make the diagnosis. Although many lipedema patients may also be overweight, those who have attempted to shed weight through diet and exercise often find that doing so is exceedingly tricky. At the same time, this is frequently not the case with standard obese women [10].

In the context of research challenges, unlike other diseases, including metabolic disorders, lipedema research has not yet fully benefited from the support of well-established imaging methods, reliable accessibility for sample collection, and appropriate animal models. Several advancements in basic and medical sciences have been made feasible by observing and experimenting with animal models [187]. Additionally, studies on animal models have dramatically improved our understanding of the biological functions and the pathology of diseases and helped translate basic research into clinical applications. No ideal animal model has been developed for the experimental study of the causes of lipedema, the lack of response to exercise, caloric restriction, or bariatric surgery, and the potential treatments. In the future, a lipedema mouse model may be used to gain insights into this complex disease. Thus, the knowledge acquired from such animal-model-based studies can be faithfully utilized to diagnose affected patients.

**Table 2 biomedicines-10-03081-t002:** Summary of genes and micro-RNAs screened and validated for lipedema.

Genes (Involved in)	Known Functi [188] on	Expression Level in Lipedema Samples (Lipoaspirates/SVF/2D ADSCs/3D Spheroids Derived from ADSCs) as Compared to Healthy Lean Controls
Synthesis of adipokines		
ADIPOQ	Encodes for adiponectin, which regulates fatty acid and glucose metabolism [189,190].	Unchanged [22,36,43]
LEP	Encodes the synthesis of leptin, which has myriad functions, mainly promoting energy expenditure and reducing food intake in a negative feedback loop [142,191].	Unchanged [22,43], Increased [44]
CFD	Encodes for adipsin, secreted by adipocytes promoting lipid accumulation and differentiation [192].	Decreased [22]
Transcriptional regulation of adipogenesis		
PPARG	Master regulator of adipogenesis and has functions ranging from lipid metabolism and promoting insulin sensitivity to adipokine secretion [193,194].	Unchanged [22,36], Increased [44]
C/EBPD	Upstream regulator of PPARG, a marker of early adipogenesis and a regulator of estrogen. Knockdown of C/EBPD decreased cell growth on induction of differentiation and prevented the accumulation of lipids in adipocytes [195]. The increased expression has also been linked to lymph angiogenesis (regulated via HIF1A) [196].	Decreased [22]
KLF4	Kruppel-like factors (KLFs) are a subfamily of the zinc finger class of DNA-binding transcriptional regulators [197]. Under physiological conditions, the phenotype of adipose tissue macrophages is M2, known to promote an inflammatory phenotype [83,197]. KLF4 has been reported to promote M2 polarization, activation of PPARG, and promote browning [197].	Decreased [22]
NCOR2	A nuclear cofactor that is recruited by PPARG in the absence of ligand and can downregulate the proadipogenic activity of PPARG [198].	Decreased [22]
ZNF423	ZNF423 is a dominant regulator of early adipocyte determination. Ectopic expression of ZNF423 activates the expression of PPARG in 3T3-L1 cells, which enables these cells to undergo differentiation [43,199,200]. In addition, expression of ZNF423 was found to be induced by estradiol in breast cancer lines, indicating a link between dysfunctional estrogen signaling pathway and lipedema [43,58,200].	Increased [43]
Modulation of insulin sensitivity		
INSR	Insulin, an endocrine hormone, plays a role in the differentiation of adipocytes as well as their metabolic function [201].	Unchanged [22]
GLUT4	Reduced GLUT4 expression in adipocytes has been associated with insulin resistance [202].	Unchanged [36,44]
Modulation of cell cycle and proliferation		
CCND1	Upregulation of cyclin D1 supported early-stage cell differentiation of 3T3-L1 cells by promoting mitotic clonal expansion (MCE) [53].	Increased [22]
CDC20	Scaffold/matrix-attachment-region-binding protein 1 (SMAR1) acts as a negative regulator for adipogenesis. CDC20 mediates proteasomal degradation of SMAR1, and its higher expression correlates with increased adipogenesis in 3T3-L1 cells [203].	Increased [24]
CENPF	It is a microtubule-binding protein, which interacts with syntaxin 4 and regulates membrane trafficking of GLUT4 in 3T3-L1 cells [204].	Increased [24]
BIRC5	Encodes for survivin, which plays a role in adipocyte homeostasis. In HFD conditions, adipocytes express the survivin gene, which inhibits DNA damage stress responses and TNF-A-induced lipolysis in response to a challenge such as obesity [188].	Increased [24]
KIF14	Adipocyte-specific deletion of KLF14 in female mice resulted in increased overall fat mass [205].	Increased [24]
BUB1	Encodes for BUB1 protein, a mitotic checkpoint serine/threonine kinase protein implicated in increased cellular proliferation by interrupting G1/S phase transition and causing increased histone H2A phosphorylation [24,143].	Increased [24]
Lipid metabolism		
LPL	In a high-fat-diet study, LPL knockout mice were found to have reduced adiposity and an improved profile for plasma insulin and adipokines [206].	Unchanged [36]
FABP4	FABP4 downregulates PPARG. Increased adipogenesis was observed in preadipocytes isolated from FABP4 null mice compared to the control [207].	Unchanged [43]
Lymphatic function		
VEGFR-3	VEGFR-3 is an inflammatory marker contributing to adipose insulin resistance by recruiting inflammatory macrophages [208]. Activation of VEGFR-3 in lymphatic endothelial cells promotes the formation of lymphangiogenesis within and around tumors and facilitates metastasis [209].	Increased [32]
PDPN	Podoplanin is predominantly expressed by lymphatic endothelium and is regulated by PROX-1. Mice lacking podoplanin displayed reduced lymphatic transport, congenital lymphedema, and cutaneous and intestinal lymphatic channel enlargement [210,211].	Unchanged [32]
PROX-1	PROX-1 is known to be the master regulator of lymphangiogenesis [210,212,213]. In microvascular endothelial cells, ectopic expression of PROX-1 upregulated the lymphatic endothelial-cell markers podoplanin and VEGFR-3 [210].	Unchanged [32]
LYVE1	LYVE-1 is a well-known lymphatic marker and is implicated in the trafficking of cells within lymphatic vessels and nodes [214]. LYVE-1-positive macrophages present in epididymal AT induce angiogenesis through secretion of MMP-9, MMP-12, and MMP-7 essential for longitudinal growth of AT [215].	Unchanged [32]
CCL21	Secreted by lymphatic endothelial cells. Essential for the guidance of antigen-presenting CCR7-positive dendritic cells from peripheral tissues to the lymphatic capillaries regulating immune response and transmigration across the blood and lymphatic endothelium [216].	Unchanged [32]
Maintaining endothelial barrier		
CDH5	The main constituent of endothelial adherens junction and marker for endothelial barrier integrity [217]. Variations in the CDH5 gene have been associated with lymphatic malformations and predisposition for lymphedema [121].	Decreased [43]
TIE-2	Serves as a receptor for angiopoietins 1 and 2. Interendothelial cell–cell adhesion and ANG1-TIE-2 signaling cooperatively regulate endothelial cell survival and structural integrity [218], and dysregulation of TIE-2 is linked to leaky vessels in lymphedema [219].	Unchanged [43], Decreased [32]
Thermogenic function		
UCP1	Involved in uncoupling respiration from ATP synthesis to generate heat in brown and beige adipocytes aiding adaptive thermogenesis [220,221,222].	Unchanged [22]
Inflammatory processes		
CD11c (ITGAX)	Acts as a marker gene for immune-cell infiltration. Plays a key role in T-cell activation and accumulation in AT, leading to insulin resistance in obesity [223].	Increased [22]
Altered micro-RNAs		
miR-16-5p	Significantly upregulated during differentiation of 3T3-L1 preadipocytes by regulating EPT1 gene expression [224]. It also plays a role in inhibiting cell proliferation through the regulation of cyclin genes, including CCND1, which have already been shown to be highly expressed in lipedema AT [22,225].	Decreased [78]
miR-29a-3p	In a 15 week weight-loss intervention study, upregulation of miR-29a-3p in SAT was observed, which also negatively correlated with lipoprotein lipase levels [226].	Decreased [78]
miR-454-3p	In bovine mammary cells, expression of miR-454-3p and PPARG was negatively correlated [142].	Decreased [78]
Hormonal regulation		
CYP19A1	CYP19A1 encodes for the aromatase enzyme, which plays an essential role in estrogen biosynthesis by chemically aromatizing androgens to estrogen (this conversion is tightly regulated, and any dysfunction is related to a range of disorders—breast cancer and polycystic ovary syndrome, to name a few) [43,227].	Increased [43]

SVF: Stromal vascular fraction; ADSCs: Adipose-derived stem cells; ADIPOQ: Adiponectin; LEP: Leptin; CFD: Complement factor D; PPARG: Peroxisome-proliferator-activated receptor gamma; C/EBPD: CCAT-enhancer-binding protein delta; KLF4: Kruppel-like factor 4, NCOR2: Nuclear receptor corepressor 2, ZNF423: Zinc finger protein 423, INSR: Insulin receptor; GLUT4: Glucose transporter type 4; CCND1: Cyclin D1; CDC20: Cell-division cycle 20; CENPF: Centromere protein F; BIRC5: Baculoviral inhibitor of apoptosis repeat containing 5; KIF14: Kinesin family member 14; BUB1: Budding uninhibited by benzimidazoles 1 homolog; LPL: Lipoprotein lipase; FABP4: Fatty-acid-binding protein 4; SMAR1: Scaffold/matrix-attachment region-binding protein 1; VEGFR-3: Vascular endothelial growth factor receptor 3; PDPN: Podoplanin; PROX-1: Prospero homeobox1; LYVE1: Lymphatic vessel endothelial hyaluran receptor; CCL21: C-C motif chemokine ligand 21; CDH5: Vascular endothelial cadherin; TIE-2: Tunica interna endothelial cell kinase 2; UCP1: Uncoupling protein 1; CYP19A1: Aromatase; EPT1: Ethanolamine phosphotransferase 1.

Complications linked to increasingly prevalent functional and cosmetic conditions can be avoided through the early diagnosis and treatment of lipedema [179]. Hence, in order to discover the most promising treatment options, thorough investigations with longer-term follow-ups are crucially required.

## Figures and Tables

**Figure 1 biomedicines-10-03081-f001:**
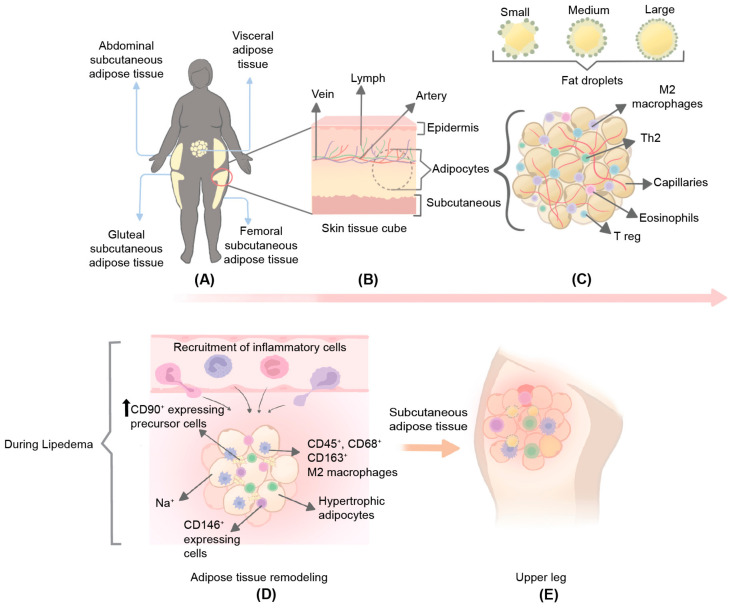
Representation of subcutaneous adipose tissue (SAT) characteristics pattern in lipedema. (**A**) The SAT location and deposition in the body; (**B**) skin tissue cube presenting their residing components; (**C**) characteristics of lean adipose tissue accompanied by ECM, immune cells, and various forms of typical fat droplets; (**D**) hypertrophy and hyperplasia in SAT with immune-cell requirements; and (**E**) inflammation, excessive fluid deposition, and swelling in the upper leg/thigh, which are the common characteristic of individuals with lipedema.

**Figure 2 biomedicines-10-03081-f002:**
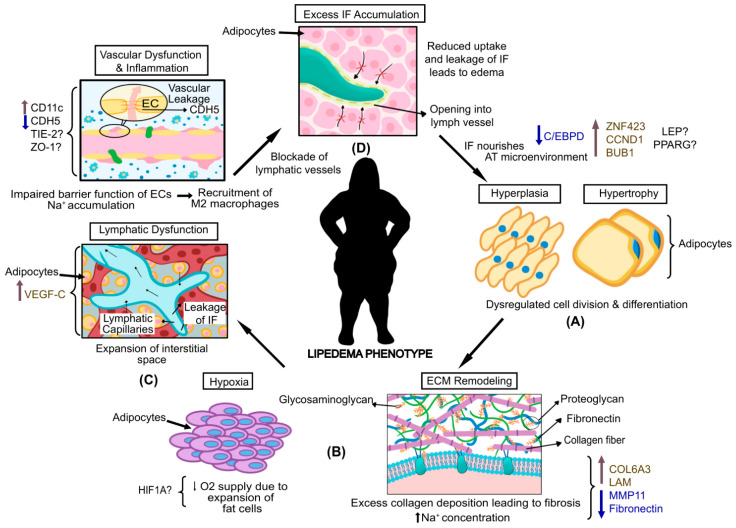
The cycle of events underpinning the lipedema phenotype. (**A**) Increased expression of genes associated with mitotic clonal expansion (MCE) and cell proliferation leads to hyperplasia. Although increased accumulation of lipids within fat cells leading to hypertrophy has been confirmed in lipedema, the expression of markers associated with such a finding is still under debate. (**B**) Due to the overgrowth of fat cells, there is a reduced oxygen supply, and extracellular matrix (ECM) remodeling is observed. ECM remodeling includes sodium (Na^+^) concentration, collagen deposition, and disturbances in the glycocalyx (including proteoglycans and glycosaminoglycan), leading to microangiopathy and fibrosis. (**C**) Endothelial permeability and paracellular leakage are increased due to the loosening of the tight junctions between endothelial cells (ECs), further deteriorating vascular impairment and causing inflammation. In addition, the ability of the lymphatic capillaries to take up interstitial fluid (IF) is diminished, causing leakage of IF, promoting lymphatic dysfunction and expansion of interstitial space. (**D**) Pathogenic changes both at the vascular and lymphatic levels contribute to the accumulation of IF, which leads to the deposition of fluid. In addition, the excess IF surrounding the fat cells acts as a source of nutrients, further contributing to the pathological expansion of the fat cells, eventually causing the remodeling of SAT and lipedema phenotype cyclically. Blue: Decreased expression of these genes observed in lipedema; Brown: Increased expression of these genes observed in lipedema; Black: Contradictory findings have been reported for these genes.

**Figure 3 biomedicines-10-03081-f003:**
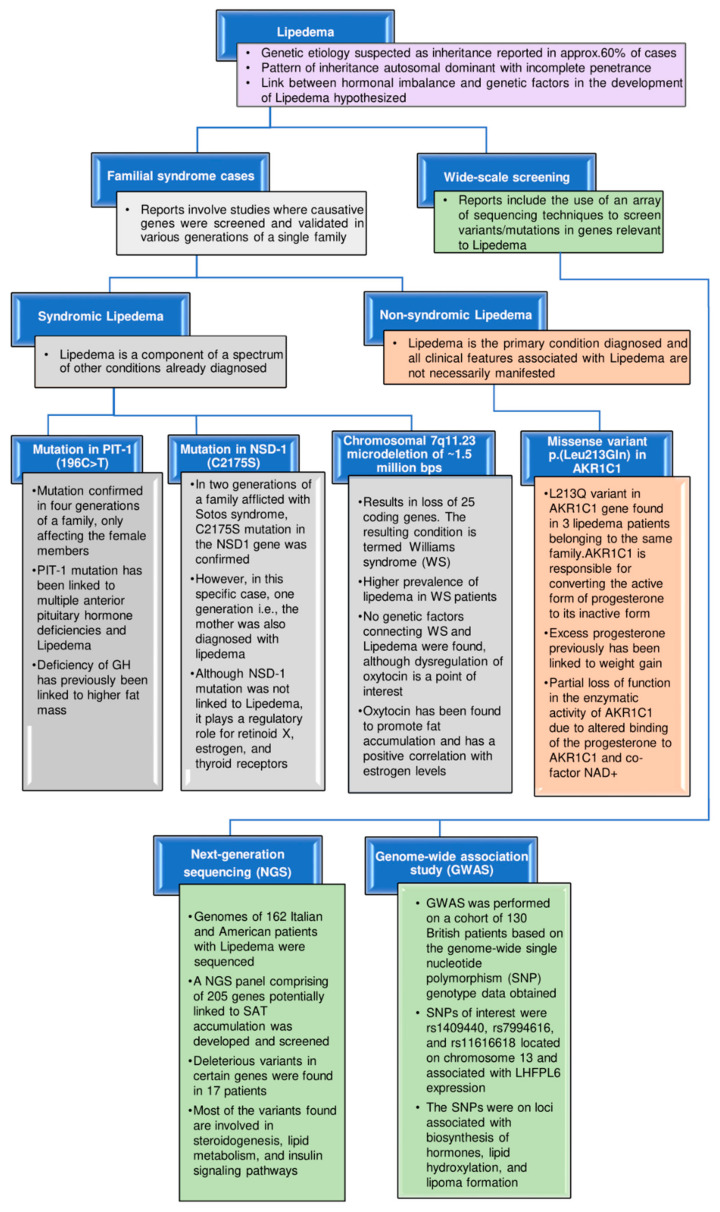
The flowchart depicts the holistic view of the current genetic implications for lipedema. Screening and validation of causative genetic factors performed in various generations of a single family are highlighted in familial syndrome cases of patients with lipedema. This is further categorized based on syndromic (described based on the chromosomal structural changes and a mutation in specific genes) and nonsyndromic (development of missense gene variants, i.e., L213Q variant in AKR1C1 gene) conditions. The wide-scale screening included next-generation sequencing (NGS) and genome-wide association studies (GWAS), revealing the variants/mutations in genes found to be linked or relevant to lipedema.

**Figure 4 biomedicines-10-03081-f004:**
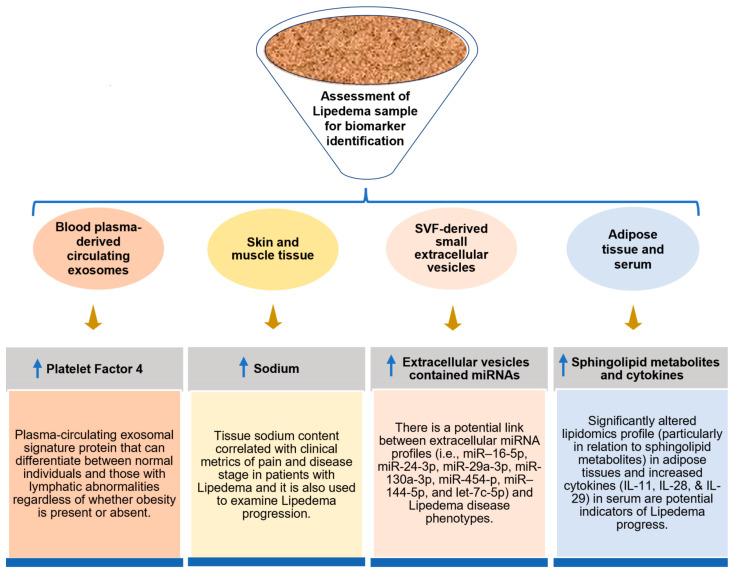
Schematic representation of identified lipedema biomarkers. This image shows the findings of some promising biomarkers from various sources of the lipedema samples and the underlying investigation to find their potential as diagnostic tools for lipedema.

**Table 1 biomedicines-10-03081-t001:** Stages of lipedema are categorized based on adipose tissue structure, mobilization pattern, and pathological conditions observed under clinical investigation of patients with lipedema.

Stage I	Stage II	Stage III	Stage IV
Smooth skin texture, enlarged subdermis, and pearl-sized nodules in a hypertrophic SAT layer. Once in a while painful and has a subdermal pebble-like feel due to underlying loose connective tissue fibrosis.	Skin depressions with pearl-to-apple large-sized masses that form in the skin and adipose tissue. Palpable nodules and bands of perilobular fascia thicken and contract. Inflamed appearance of skin due to progressed fibrotic changes, pulling the skin down in the mattress pattern due to excess tissue.	Patients feature more painful, increased lipedema tissue that is more fibrotic in texture, with numerous large subdermal nodules. Skin thins and loses elasticity, allowing SAT to grow excessively and fold over, inhibiting flow.	It is characterized by lipolymphedema (concomitant lymphedema with lipedema). Features large overhangs of fat tissue on legs or arms and large fat tissue extrusion on the legs.

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
