# Peer review of "Lipedema: Insights into Morphology, Pathophysiology, and Challenges"

_biomedicines, 2022, doi:10.3390/biomedicines10123081_

Round 1

Reviewer 1 Report

It is a great and much needed review article covering all the major findings in the Lipedema research field. The authors did a good job in compiling all the data presented in the Literature. However, the article lacks structural organization.

Comments:

1.       The title needs to be changed to reflect the content of the review.

2.       No subtitle is needed in the introduction

3.       For the morphology of Lipedema (2.1), the authors started by describing the stem cells in the thigh tissue. It would be better to describe the Lipedema adipose tissue (2.2) morphology first and then dive into the different cell types in tissues. Additionally, it would be good to start with morphology of the skin of patients as they describe the different stages of Lipedema.

4.       The authors need to add reference to line 63 and 112.

5.       Line, 113, The excess of IF should go under morphology or ECM remolding effect. The author should remove the word: “some” from the title. I would suggest compiling 2.3 and 2.4 in one paragraph.

6.       Lymphatic function and vascular function should go under different title and maybe combined with the Lymphatic system (3.5, line 488), maybe Vasculature in Lipedema AT.

7.       Connective tissue and ECM remodeling paragraphs should be combined with 3.3 paragraph. The authors should highlight the fact that Lipedema is connective tissue disorder.

8.       Mechanisms Implicated in the Pathogenesis of Lipedema should be combined with the pathophysiology

9.       Line 215, the paragraph “Altered gene expression of adipogenic and hormonal markers” should be divided into subparagraphs including fibrosis and inflammation or shortened. The Omics data can be moved to the genetic implications (4, line 627). There is a lot of overlapping between the paragraphs. Please re-write and be concise.

10.  The figures presented are very well illustrated, however, adding Key indicating the different cell types will help readers to navigate through figure. For example, in Figure 3, A, the cells are adipocytes, and What the cell types are in B, the purple cells? And same the in D? Figure 2 can be a table

11.   The authors should have a paragraph that describes the different stages of Lipedema the symptoms and potential treatment. This paragraph should not be under ECM remolding.

12.   Biomarker paragraph needs to be shortened and more specific to Lipedema findings.

13.   Technology advancement describes in here are techniques and should be rewritten.

14.   The authors talked about Lymphoscintigraphy (line 987) in conclusion and that should be under diagnostic tools.

15.   The authors did not mention any clinical studies for lipedema patients which will support the lymphatic and vascular findings.

Reviewer 2 Report

1.     The overall impression is that, as lots of facts are being brought up, the sentences are sometimes too wordy, and many paragraphs are not very well structured, making the reading hard to follow. In other words, it would be nice to structure the text around a thread the reader can follow, and support it with references. What I would like to see in a review is more discussion, and less of the “list of facts” structure. I would recommend the authors to go through the whole text and at least try to improve the writing style a bit. Alternatively, maybe there’s an option of having a professional editor work on the text? And I don’t mean this to be excruciatingly critical, because I do like the content of the review; it’s just the writing style needs some good polishing. For example see the sentence starting at line 718 – it includes the word “lipedema” three times (!). Why not make it more concise? E.g. “SNP genotyping of white British lipedema patients revealed strong genetic link to the disorder, with loci associated with hormone biosynthesis, lipid hydroxylation, and lipoma formation.”

2.     Also, I have some more specific comments below.

3.     Is there are a reason to capitalize the word “Lipedema”? If not, please use lower-case “lipedema”.

4.     Line 67-70, the description of lipidema SAT – sounds like the features overlap a lot with obese SAT. What are the differences between lipidema and obese SAT? The quantities of fibrosis, hypertrophy, macrophages? Or other features?

5.     line 127, it sounds like the legs delayed radionuclide transit, but I’m sure it’s the veins. Please rewrite the sentence.

6.     Lines 142-145, the sentence should be rewritten (two sentences perhaps?), as the message is too wordy. I also don’t follow how loss of elasticity in adipose tissue leads to hypermobile joints – please clarify.

7.     Chapter 2.3.4 ECM remodeling – what is known about the level of hyaluronic acid, which also binds lots of water. And could lipidema then be treated with hyaluronidase and/or other deglycosylation, and would a local (non-systemic) treatment be possible? What about treatment with Imatinib which has been reported to in part normalize interstitial fluid flow and extracellular matrix?

8.     I also find the section 2.4 a bit messy to read, can this paragraph be rewritten with better structure and flow?

9.     in section 3.1 – is anything known about the insulin sensitivity of lipidema adipocytes?

10. Line 366 mentions a study that combined immunostaining with machine learning, but there’s not much mentioned about the outcome of the machine learning part, or?

11. Line 372 first word should be “occludens” not “occuludens”

12. Line 413, “except for MMP11 which showed a decreased expression (not significant)” – not sure this is worth mentioning, since the finding was not significant?

13. line 509 it sounds like the subjects were opposing the prior report, please rewrite the sentence.

14. line 595 “causes issues in the normal functioning” – instead of the slang-like “issues” perhaps use “problems” or “disrupts”

15. line 611, “to avoiding”, change to “to prevent”

16. line 999, remove the comma after “while,”

17. In section 4, have the genetics studies been replicated in other cohorts? Are there any functional genomics studies that support the genetics observational data, in particular for AKR1C1? Are there any in vitro models for studying lipedema, where the genome could be modified by eg CRISPR?

18. Do the genetics studies (section 4) point to any coherent molecular pathway(s) that could shed some light on lipedema etiology? Or maybe there are specific stages of lipidema wherein the different genes are differentially expressed?

19. Can the authors suggest any precision medicine treatment of lipidema, tailored to the specific features of the disorder, or to genetics?

Round 2

Reviewer 1 Report

None

Author Response

We thank the reviewer for his time and suggestions for the improvement of our manuscript.

We have now applied the Grammarly Premium which reviewed our spelling, grammar, punctuation, clarity, engagement, and delivery mistakes in the text. 

Reviewer 2 Report

It looks better now. I don't know if my pdf file is 'corrupted' but I cannot read some of the lines (eg line 1321, 1465, and some other). I assume that there will be another check up on the text before publishing?

Some issues I spotted that need to be fixed:

Please look over lines 100-104 - there's a duplicated sentence there.

Lines 1168, 1187 and 1236 are the same paragraphs.

Repeated paragraph headers on page 33 (Lipid profile, cytokines)

repeated paragraphs starting on lines 1430 , 1466 and 1482

Author Response

We thank the reviewer for his time and suggestions for the improvement of our manuscript. It seems the uploaded PDF file was somehow corrupted and had an issue.

We have now applied the Grammarly Premium which reviewed our spelling, grammar, punctuation, clarity, engagement, and delivery mistakes in the text.